# Digital Inclusive Finance, Environmental Regulation, and Regional Economic Growth: An Empirical Study Based on Spatial Spillover Effect and Panel Threshold Effect

**Rijia Ding, Fenfen Shi * and Suli Hao**

School of Management, China University of Mining and Technology (Beijing), Beijing 100083, China; dingrijia@cumtb.edu.cn (R.D.); 108934@cumtb.edu.cn (S.H.)
* Correspondence: bqt2000503019@student.cumtb.edu.cn

**Abstract:** The development of digital financial inclusion has added a new vitality to economic growth, and environmental regulation is an important tool to achieve sustainable economic growth. Therefore, whether there is a synergistic effect between these two factors of economic growth is a topic worth exploring. This paper uses the space econometric model and threshold model to explore the impact of digital financial inclusion and environmental regulation on regional economic growth using panel data from 30 Chinese provinces, collected between 2011 and 2019. The research results prove that the development of digital financial inclusion and the improvement in the intensity of environmental regulation have a significant direct promotion effect and negative spatial spillover effect on regional economic growth. Moreover, the two have a significant synergistic effect on regional economic growth. A panel threshold analysis showed that, with the improvement in the level of digital financial inclusion, the regression coefficient of environmental regulation changed from negative to positive, which played a significant role in promoting regional economic growth. The heterogeneity analysis found that digital inclusive finance in eastern regions of China plays a greater role in promoting the economy, whereas environmental regulation in the central region plays a greater role in promoting the economy. The synergy between the two in the central region greatly promotes economic development. When digital inclusive finance is used as the threshold variable, environmental regulation in eastern and western regions has a single-threshold effect on regional economic development. Based upon these research results, this paper proposes that a coordination mechanism between digital financial inclusion and environmental regulation should be established to give full play to their synergies in sustainable economic growth.

**Keywords:** digital financial inclusion; environmental regulation; economic growth; spatial spillover; threshold effect





## 1. Introduction

Since the conception of financial inclusion by the United Nations, it has become a priority policy option for many countries to address financial exclusion. At present, the global practice of inclusive finance has been used for more than ten years. In addition, it has completed the development process of "small and micro finance-internet finance-digital inclusive finance". It has significantly contributed to global financial equity and sustainable development [1]. Digital inclusive finance has become a new idea for the development of inclusive finance and a hotspot of innovation in the financial field, which is in line with the requirements of the era of digital intelligence. Currently, the wave of digitalization has considerably affected all areas of the traditional economy. In addition, coupled with the sudden outbreak of the COVID-19 epidemic, the financial industry has accelerated its transformation to digitalization. This has brought many benefits to opening up the "last mile" of inclusive finance, which provides the possibility for the rapid development of China's economy [2]. Therefore, digital financial inclusion is considered an important driving force

for economic growth. Additionally, environmental regulation is considered an important assurance for promoting sustainable and healthy economic development. In terms of total pollutant emissions, China ranks among the top in the world, showing the characteristics of a low resource utilization efficiency and huge potential for improving environmental efficiency. Therefore, under the rigid constraints of resources and the environment, it is essential to accelerate the transformation of the development mode. For economic transformation and development, environmental regulation has become an important driving force. It encourages enterprises to change their original production modes, strengthen technological innovation, and improve the utilization efficiency of resource elements and environmental efficiency, which requires enterprises to have sufficient funds in order to ensure technological innovation. However, digital inclusive finance can reduce transaction costs and financial service thresholds, improve capital allocation and industrial financing efficiency, and effectively complement the financial system of enterprises by providing financial products or services in a digital form, which can encourage enterprises to achieve green transformation and sustainable economic growth.

Digital financial inclusion and environmental regulation both play an important role in China's economic growth. Therefore, the following questions are worth exploring. What is the role of China's digital financial inclusion and environmental regulation in the process of regional economic growth? What are the spatial effects and regional heterogeneity? What is the internal mechanism of action of the three factors? To answer the above questions, this paper uses the spatial econometric model and the panel threshold model to analyze the impact of digital financial inclusion and environmental regulation on regional economic growth. This study aims to provide a quantitative basis and decision support for government departments to formulate digital financial inclusion development policies and environmental regulation policies in economic development.

This study's main contributions are as follows: First, this paper incorporates digital financial inclusion, environmental regulation and regional economic growth into the same research framework and analyzes their mechanisms of action, which enriches the literature on digital finance and environmental regulation. Second, the index method is used to construct a comprehensive index of environmental regulation intensity, and three spatial econometric models are used to analyze the independent effects and synergistic effects of digital financial inclusion and environmental regulation on regional economic growth, laying a foundation for empirical research. Third, the panel threshold model verifies that the impact of environmental regulation on regional economic growth is affected by digital financial inclusion, which further illustrates the relationship between digital financial inclusion, environmental regulation and regional economic growth. The above research provides a basis for government departments to formulate environmental regulation policies that are compatible with the development level of digital financial inclusion.

The remainder of this paper is arranged as follows. Section 2 is devoted to a literature review. Section 3 is the theoretical mechanism and research hypothesis. Section 4 explains the meaning of the indicators, describes the data and introduces the model. Section 5 conducts an empirical analysis of the spatial econometric model and the panel threshold model. Section 6 presents research conclusions, policy recommendations and limitations of the research.

## 2. Literature Review

Many studies have verified the influence of inclusive finance development on economic growth, and most scholars believe that the development of inclusive finance can promote economic growth. The result of digital financial inclusion can not only promote economic growth but also effectively improve the stability of the financial system and provide increasingly extensive opportunities for society, which helps improve the living standards and welfare of low-income groups, thereby continuously promoting economic growth [3–6]. In a cross-country study, Yan et al. concluded that digital financial inclusion has a significant positive impact on economic growth and has spatial spillover effects into

neighboring countries [7]. Myovella et al. proved that digitalization positively contributed to economic growth either in Sub-Saharan Africa (SSA) or in OECD economies [8]. Since the concept of digital financial inclusion was introduced into China, scholars have conducted a lot of theoretical and practical research on the impact of digital financial inclusion on economic growth. Most scholars believe that the development of digital inclusive finance contributes to economic growth. For example, Hao et al., Lv et al. and Yang et al. concluded that digital financial inclusion has a significant positive effect on economic growth, but there are slight differences in the spatial correlation and promotion effect among regions [9–11]. Jiang et al. also made similar conclusions [12]. Studies by Wang et al. concluded that digital finance palys a significant role in promoting economic growth both on an overall level and from the three dimensions of coverage, depth of use, and digitalization [13]. Yang et al. and Chu et al. proposed that digital inclusive finance has a positive direct effect on local economic growth and a negative spatial spillover effect on surrounding areas [14,15]. However, some scholars believe that there is a nonlinear relationship between digital financial inclusion and regional economic growth. For example, He et al. pointed out that there is a threshold effect between the development of digital inclusive finance and economic growth. When the development level of digital financial inclusion exceeds this threshold, it can further stimulate its positive effect on economic growth [16]. The research by Yang et al. shows that the impact of digital financial inclusion development on economic growth has a significant Internet threshold effect [17]. Zhan pointed out that digital financial inclusion has an inhibitory effect on the quantity of economic growth, but it will significantly promote the quality of economic growth. There are U-shaped and inverted U-shaped relationships between digital financial inclusion and the quantity and quality of economic growth, respectively [18].

In order to mediate environmental externalities and promote green and sustainable development, environmental regulation is considered an important starting point for the government. The current studies by research scholars are primarily divided into three points of view in the study of environmental regulation and economic growth. According to the first view, environmental regulation can significantly promote economic growth. Reasonable environmental regulation can promote corporate innovation, reduce corporate costs, improve profitability, and promote economic growth based on the "Porter hypothesis" [19,20]. Peng explained the innovation effect brought by environmental regulation and verified that the innovation compensation effect can improve the productivity of enterprises [21]. In China's economic transformation, Wang and others also believed that environmental regulation could accelerate the technological innovation of enterprises, improve energy efficiency, and promote regional economic growth [22]. He and Hu et al. supported the "Porter Hypothesis" [23,24]. According to the second view, environmental regulation shows a negative effect on regional economic growth. Environmental regulation increases the cost of enterprises and shows a crowding-out effect on production based on the explanation of the "cost effect", which reduces the profitability of enterprises, thereby inhibiting economic growth. Jorgenson et al. and Gray et al. successively studied the impact of environmental regulation in pollution-intensive industries and manufacturing sectors in the USA, observing that environmental regulation shows a significantly negative effect on economic growth [25,26]. Chong et al. revealed that environmental regulation has remarkable negative effects on economic growth in China [27]. This conclusion is also confirmed by Mi et al. [28]. According to the third view, there is no significant causal relationship between environmental regulation and regional economic growth, which indicates that the impact of environmental regulation on technological innovation or enterprise performance is uncertain [29,30]. Considering the combined effect of the "Porter hypothesis" and the "cost effect", research scholars have observed that there is a nonlinear relationship between environmental regulation and regional economic growth [31,32]. Xiong verified the existence of a U-shape between environmental regulation and economic growth based on provincial panel data from 2004 to 2008 [33]. However, Cao et al. verified the inverse U-shape between environmental regulation and economic growth using the Yangtze River

Delta region as a sample [34]. Some scholars concluded that environmental regulation is affected by human capital and entrepreneurial level in the process of affecting economic growth, showing a nonlinear relationship [35,36].

Furthermore, scholars also conducted research into the joint role of financial development and environmental regulation, mainly focusing on green total factor productivity, green development efficiency, and industrial structure upgrading. For example, Ni et al. confirmed that the independent effects of financial development and environmental regulation have a certain role in promoting green total factor productivity. However, the combination of financial development and environmental regulation inhibits green total factor productivity [37]. Li et al. pointed out that the strengthening of environmental regulation can promote the allocation of financial resources to the secondary industry and reduce the efficiency of green development [38]. Li et al. confirmed that the intersection of environmental regulation and financial development can effectively promote the upgrading of industrial structure [39]. Wang et al. proposed the opposite conclusion [40]. With the development of digital financial inclusion, scholars have also carried out preliminary research on the joint role of digital financial inclusion and environmental regulation. Shangguan et al. proposed that the interaction between digital finance and environmental regulation can promote green total factor productivity [41]. Li et al. pointed out that digital inclusive finance and environmental regulation play a positive role in promoting the industrial structure, and digital inclusive finance is an important moderating variable for environmental regulation to affect the upgrading of industrial structure [42]. Cao et al. pointed out that financial supervision and environmental regulation from the Chinese government can reinforce the role of digital finance in promoting energy–environmental performance [43].

Based on the literature review, we can find some characteristics of previous studies. First, previous research mainly focused on the relationship between digital financial inclusion and regional economic growth, and the relationship between environmental regulation and regional economic growth. Second, some scholars discussed the intersection of financial development and environmental regulation, but they have come to different conclusions. In addition, with the development of digital financial inclusion, scholars have gradually paid attention to related research on digital financial inclusion and environmental regulation, mainly focusing on research on green total factor productivity, industrial structure upgrading, and energy-environmental performance. The existing literature provides theoretical and empirical support for this research. Based on the existing research, this paper further enriches the theoretical and empirical research on digital financial inclusion, environmental regulation and regional economic growth. Mainly mentioned in the following three aspects. First, the mechanism of interaction between digital financial inclusion, environmental regulation and regional economic growth is discussed. Second, a comprehensive index of environmental regulation is constructed, and the spatial Durbin model is used to test the independent and synergistic effects of digital financial inclusion and environmental regulation on regional economic growth. Third, the panel threshold model is used to explore the impact of the intensity of environmental regulation on the regional economy under different development levels of digital financial inclusion.

## 3. Theoretical Mechanisms and Theoretical Hypotheses

Digital financial inclusion is the product of the combination of financial inclusion and Internet technology. It is an effective supplement to traditional finance. First, digital inclusive finance can lower the threshold of financial services and provide financial services to users in the region at the micro level, which alleviates financial exclusion to a certain extent and activates the local economy. Secondly, digital inclusive finance can effectively break geographical restrictions and enable remote areas to enjoy financial services, which can improve the efficiency of such services, stimulate local market vitality and increase employment opportunities. Finally, digital financial inclusion can use information technology to reduce the degree of information asymmetry, ease the financing constraints of enterprises, and activate the innovation vitality of enterprises, which can promote indus-

trial transformation and achieve economic growth. However, regions with a high level of digital service development may absorb customer resources from other regions with their advanced technologies and high-quality services, resulting in negative spatial spillover effects. In summary, the following assumptions are proposed in this paper:

**Hypothesis 1 (H1).** *The development of digital financial inclusion promotes regional economic growth and shows negative spatial spillover effects.*

Environmental regulation is an important starting point for the Government to mediate environmental externalities and promote sustainable development. Environmental regulation can raise the entry threshold of enterprises, eliminate or transfer high-polluting enterprises and promote the industrial rationalization process of enterprises [44]. Stricter environmental regulations can curb the short-sighted behavior of enterprises, promote the technological innovation of enterprises and encourage more enterprises to upgrade elements in the process of "learning by doing". This will help promote the upgrading of the industrial structure [45]. Moreover, driven by technological progress, low-productivity sectors gradually withdraw from the market, and high-productivity sectors continue to refine the division of labor, which can effectively promote economic growth. Furthermore, according to the "pollution paradise hypothesis", when the environmental regulation of the economically developed province is stronger, it will seek to optimize the industrial structure. At this time, pollution-intensive industries are aimed toward other areas with loose environmental regulations [46], resulting in negative spatial spillover effects. Based on this fact, the following assumptions are proposed in this paper:

**Hypothesis 2 (H2).** *The increase in the intensity of environmental regulation promotes economic growth in the region and shows a negative spatial spillover effect.*

The sustainable development of the economy requires a considerable environmental investment and reallocation of capital by industries with the collective efforts of both the environmental protection department and financial department. Then, there may be a process of mutual adjustment between environmental regulation and digital finance. On the one hand, environmental investment generally has the characteristics of high risks, low early-stage returns and long cycles. In tightening environmental regulations, various types of enterprises are facing constraints on production funds. However, digital finance can effectively achieve accurate data matching and more accurate risk assessments that rely on information processing methods. It can also use its information and technological advantages to broaden corporate financing channels, reduce rent-seeking space, alleviate financial discrimination [47], and promote enterprises' technological innovation. On the other hand, when a company faces looser capital constraints, its cash flow pressure is also less. Compared with reduced production, the benefits of corporate pollution control may be greater. At this time, companies will adopt an environmental investment and pollution control methods to reduce emissions [48]. In addition, as the intensity of environmental regulation increases, it can effectively restrict the flow of resources to high-polluting enterprises, reasonably guide the flow of funds into green industries [49], and improve the enthusiasm of enterprises for green technology activities by using the cross-integration and innovation of financial products or services, which can fundamentally solve the problem of environmental pollution in order to achieve sustainable economic growth [50]. The framework shown in Figure 1 can be obtained by considering the above studies on the impact of digital financial inclusion, environmental regulation, and regional economic growth. Based on this fact, the following assumptions are proposed in this paper:

**Hypothesis 3 (H3).** *Digital financial inclusion and environmental regulation can synergistically promote regional economic growth.*

**Hypothesis 4a (H4a).** *The effect of digital financial inclusion development on economic growth is affected by the intensity of environmental regulation.*

**Hypothesis 4b (H4b).** *The effect of the intensity of environmental regulation on economic growth is affected by the development of digital financial inclusion.*

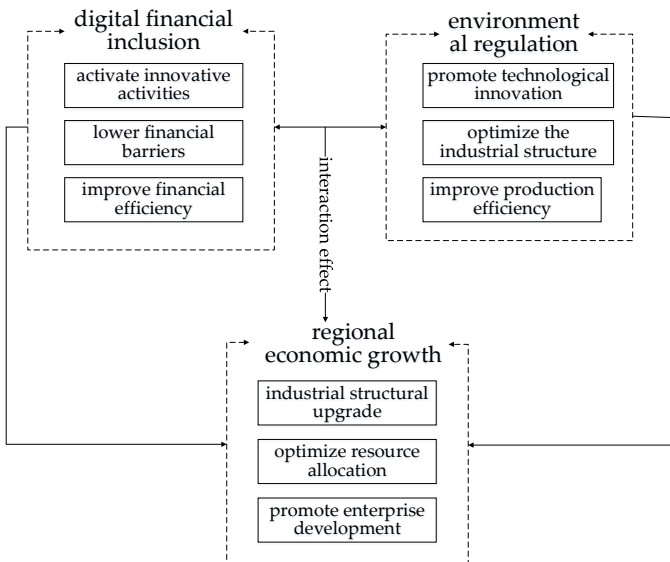

**Figure 1.** Theoretical framework of digital financial inclusion, environmental regulation and regional economic growth.

## 4. Data Sources and Research Methods

### 4.1. Index Selection

4.1.1. Explained Variable

Existing studies mostly use regional GDP or regional per capita GDP to represent regional economic growth. In this paper, the logarithm of regional GDP per capita was used to measure regional economic growth [51].

4.1.2. Core Explanatory Variable

In this paper, the digital financial inclusion index (divided by 100) released by Peking University is used to measure the level of digital financial inclusion based on the common practice of existing research [52]. The method of environmental regulation in this paper is slightly improved on the basis of the measurement methods of Yuan et al., Pei et al. and Song et al. [53–55]. It uses wastewater, sulfur dioxide, nitrogen oxides, smoke and dust emissions and solid-waste generation as the basic indicators to construct a comprehensive measurement index system for environmental regulation. The first step is to standardize the five individual indicators. The formula is as follows:

$$E_{ij}^s = [(E_{ij} - \min(E_{ij})] / [\max(E_{ij}) - \min(E_{ij})] \tag{1}$$

In the formula, $i$ and $j$ represent the province ($i = 1, 2, \cdots, 30$) and various types of pollutants ($j = 1, 2, \cdots, 5$). $E_{ij}$ represents the emissions of various types of pollutants in each province.

The second step is to calculate the adjustment coefficient of each individual indicator to eliminate the large pollution difference between different provinces. The calculation formula is as follows:

$$\tau_{ij} = (\frac{E_{ij}}{\sum E_{ij}}) / (\frac{Y_i}{\sum Y_i}) \tag{2}$$

$\tau_{ij}$ represents the adjustment coefficient of pollutant $j$ in area $i$, $Y_i$ represents the industrial added value of area $i$ and $\sum Y_i$ represents the national industrial added value.

The final step is to calculate the total environmental regulation intensity of each region by formula (3). In the formula, the larger the *er*, the larger the environmental regulation.

$$er = 1/\sum_{j=1}^{5} \tau_{ij} \times E_{ij}^s \tag{3}$$

4.1.3. Control Variables

Technological innovation: The number of applications for invention patents can better reflect the current technological innovation achievements of a region. Therefore, scientific and technological innovation is measured by the number of applications for invention patents [52].

Industrial structure: This is measured by the ratio of the added value of the secondary and tertiary industries to the regional GDP [56].

Human capital: This is measured by the proportion of the number of students in colleges and universities in the resident population [57].

Urbanization: This is expressed as the proportion of urban population in the total population [58].

The degree of government intervention: Government fiscal spending may impact economic growth by affecting resource allocation [59], The degree of government intervention is measured by the ratio of the general public budget expenditure of each regional government to the regional GDP.

Foreign direct investment: The capital accumulation and technology spillover effects brought by foreign direct investments can effectively promote the upgrading of industrial structure and promote regional economic growth. Therefore, foreign direct investment is measured by the ratio of the actual use of foreign direct investment to the regional GDP. In the calculation process, the actual utilization of foreign direct investment is converted according to the annual average exchange rate of RMB against the US dollar [60].

*4.2. Data Description*

To ensure the integrity of the data, this paper uses panel data from 30 provinces and autonomous regions in China (excluding Tibet, Hong Kong, Taiwan, and Macao) from 2011 to 2019 as the research object. Furthermore, the data of digital financial inclusion comes from the Peking University Digital Financial Inclusion Index (2011–2020). Regional economic growth, individual indicators of environmental regulation and other control variables are selected from the "China Statistical Yearbook", "China Environment Yearbook", and the statistical yearbooks of all provinces and cities in China (2012–2020). The descriptive statistics for the variables used in this study are presented in Table 1.

**Table 1.** Descriptive statistics.

| Variable | Definition | Obs | Mean | Std. Dev. | Min | Max | Unit |
|---|---|---|---|---|---|---|---|
| lnpgdp | Regional economic growth | 270 | 10.8081 | 0.4348 | 9.7058 | 12.0090 | CNY |
| df | Digital financial inclusion | 270 | 2.0336 | 0.9157 | 0.1833 | 4.1028 | – |
| er | Environmental regulation | 270 | 1.8269 | 4.0204 | 0.0617 | 24.5849 | – |
| te | Technological innovation | 270 | 3.1122 | 4.1180 | 0.0204 | 21.6469 | 10,000 pieces |
| is | Industrial structure | 270 | 0.9028 | 0.0512 | 0.7387 | 0.9973 | % |
| hc | Human capital | 270 | 0.0195 | 0.0049 | 0.0080 | 0.0345 | % |
| urb | Urbanization | 270 | 57.5142 | 12.4075 | 17.8100 | 89.6000 | % |
| gov | The degree of government intervention | 270 | 0.2490 | 0.1027 | 0.1103 | 0.6284 | % |
| fdi | Foreign direct investment | 270 | 0.0215 | 0.0166 | 0.0000 | 0.0796 | % |

*4.3. Research Methods*

4.3.1. Spatial Correlation Test

Before the introduction of the spatial econometric model, it is necessary to measure the spatial dependence of digital financial inclusion, environmental regulation and regional economic growth in China. This paper conducts analyses from two perspectives of global space autocorrelation and local space autocorrelation. The global space autocorrelation is tested by Moran's index and the calculation formula is as follows.

$$I = \frac{\sum\limits_{i=1}^{n} \sum\limits_{j=1}^{n} W_{ij}(x_i - \bar{x})(x_j - \bar{x})}{S^2 \sum\limits_{i=1}^{n} \sum\limits_{j=1}^{n} W_{ij}} \tag{4}$$

In the formula, $S^2 = \frac{\sum\limits_{i=1}^{n}(x_i - \bar{x})}{n}$ is the sample variance, $\bar{x} = \frac{\sum\limits_{i=1}^{n} x_i}{n}$; $x_i$ and $x_j$ are observations in regions; $i$ and $j$ are the total number of provinces, $W_{ij}$ is the space matrix; and Moran's I is generally in the range of $[-1, 1]$. When this latter value is less than 0, it means that the space is negatively correlated. When it is equal to 0, it means that the space is not correlated. When it is greater than 0, it means that the space is positively correlated. The local spatial autocorrelation is represented by the Moran index scatterplot. This paper adopts the economic distance matrix for three main reasons. First, the economic distance matrix can reflect areas that are not geographically adjacent but have close economic ties [57]. Secondly, the economic distance matrix can reflect the provinces with a higher degree of economic closeness in the adjacent regions. For example, Hebei Province is geographically adjacent to Beijing, Tianjin, Shanxi, Henan and other regions, but Hebei Province has closer economic ties with Beijing and Tianjin than other neighboring provinces [61]. In addition, this article refers to other scholars' research on digital financial inclusion or economics. For example, Hao et al. and Xu adopted the economic distance matrix [9,62]. Based on the above analysis, this paper constructs the economic weight matrix, the formula for this is as follows:

$$W = \begin{cases} \frac{1}{\frac{1}{n}\left|\sum\limits_{2011}^{2019} PGDP_i - \sum\limits_{2011}^{2019} PGDP_j\right|} & ,i \neq j \\ 0, i = j \end{cases} \tag{5}$$

*PGDP* represents the per capita GDP of the region and *n* is the year. This paper normalizes the economic distance space matrix.

4.3.2. Model Selection and Construction

Commonly used spatial economic models include spatial autoregression (SAR), the spatial error model (SEM), and the spatial Durbin model (SDM). SAR adds the lag term of the explained variables to the classical regression model. It is generally used to study the influence of the behavior of adjacent areas on other areas within a region. In the context of this paper, the resulting model is as follows:

$$\ln pgdp_{it} = c + \rho \times W \ln pgdp_{it} + \alpha_1 \times df_{it} + \alpha_2 \times er_{it} + \alpha_3 \times X_{it} + \varepsilon_{it} \tag{6}$$

SEM takes into account a spatial-disturbance error term based on the classical regression model and presents the influence of changes of the explanatory variables in adjacent areas on the explained variables. The model is as follows:

$$\ln pgdp_{it} = c + \beta_1 \times df_{it} + \beta_2 \times er_{it} + \beta_3 \times X_{it} + \varepsilon_{it} \tag{7}$$

The spatial Durbin model (SDM) comprehensively considers the spatial lag factors of explanatory variables and explained variables. The SDM in this study is set as follows:

$$\ln pgdp_{it} = c + \rho \times W \ln pgdp_{it} + \gamma_1 \times df_{it} + \gamma_2 \times er_{it} + \gamma_3 \times X_{it} + \\ \theta_1 \times Wdf_{it} + \theta_2 \times Wer_{it} + WX_{it} + \varepsilon_{it} \tag{8}$$

In the formula, $\ln pgdp$, $df_{it}$, and $er_{it}$ represent economic growth, digital financial inclusion and environmental regulation in region $i$ in year $t$. $X_{it}$ represents the control variable, $W$ represents the spatial weight matrix, and $\varepsilon_{it}$ represents a random disturbance term.

4.3.3. Panel Threshold Model

In this paper, the panel threshold model proposed by Hansen is selected to explore whether the explanatory variables are disturbed by the threshold effect, and the threshold model was developed with the digital financial inclusion and environmental regulation as the threshold variables. The specific form of these is shown in Equations (9) and (10):

$$\ln pgdp_{it} = C + \alpha_1 er_{it} \times I(df_{it} \leq \delta) + \alpha_2 er_{it} \times I(\delta \leq df_{it}) + \beta \times X_{it} + \varepsilon_{it} \tag{9}$$

$$\ln pgdp_{it} = C + \alpha_1 df_{it} \times I(er_{it} \leq \delta) + \alpha_2 df_{it} \times I(\delta \leq er_{it}) + \beta \times X_{it} + \varepsilon_{it} \tag{10}$$

In these formulas, $C$ represents individual effect, $\alpha$ is for the parameter of the threshold-dependent variable to be estimated, $I(\cdot)$ represents an indicator function with a value of 0 or 1, $X_{it}$ stands for the control variable, $\beta$ represents the parameter to be estimated by the control variable, $\varepsilon_{it}$ is the error term, and $\delta$ stands for the threshold value. The above formula is a single-threshold variable model, and the double-threshold model can be extended.

## 5. Empirical Results

### 5.1. Spatial Econometric Model

5.1.1. Spatial Correlation Test

Table 2 plots the global Moran's I for regional economic growth, digital financial inclusion and environmental regulation for each year, all of which are highly statistically significant. As can be seen, the spatial distribution of the regional economic growth, digital financial inclusion and environmental regulation has a significant spatial agglomeration effects and spatial spillover effects.

**Table 2.** Moran's I index of regional economic growth, digital financial inclusion and environmental regulation from 2011 to 2019.

| Year | Regional Economic Growth | | | Digital Financial Inclusion | | | Environmental Regulation | | |
|---|---|---|---|---|---|---|---|---|---|
| | Moran's I | Z Value | p Value | Moran's I | Z Value | p Value | Moran's I | Z Value | p Value |
| 2011 | 0.533 | 5.380 | 0.000 | 0.409 | 4.225 | 0.000 | 0.263 | 3.490 | 0.000 |
| 2012 | 0.533 | 5.375 | 0.000 | 0.414 | 4.321 | 0.000 | 0.260 | 3.528 | 0.000 |
| 2013 | 0.537 | 5.401 | 0.000 | 0.401 | 4.215 | 0.000 | 0.241 | 3.545 | 0.000 |
| 2014 | 0.529 | 5.334 | 0.000 | 0.431 | 4.512 | 0.000 | 0.220 | 3.851 | 0.000 |
| 2015 | 0.546 | 5.477 | 0.000 | 0.440 | 4.605 | 0.000 | 0.214 | 3.809 | 0.000 |
| 2016 | 0.540 | 5.440 | 0.000 | 0.426 | 4.475 | 0.000 | 0.238 | 3.373 | 0.000 |
| 2017 | 0.513 | 5.178 | 0.000 | 0.364 | 3.877 | 0.000 | 0.351 | 4.198 | 0.000 |
| 2018 | 0.512 | 5.192 | 0.000 | 0.318 | 3.396 | 0.000 | 0.360 | 4.264 | 0.000 |
| 2019 | 0.408 | 4.226 | 0.000 | 0.322 | 0.322 | 0.000 | 0.390 | 4.637 | 0.000 |

This study selected two cross sections of time: 2011 and 2019. The spatial clustering characteristics of regional economic growth, digital financial inclusion and environmental regulation in 30 Chinese provinces are analyzed using the Moran scatter plots (Figures 2–4) and the natural fracture method (Figures 5–7).

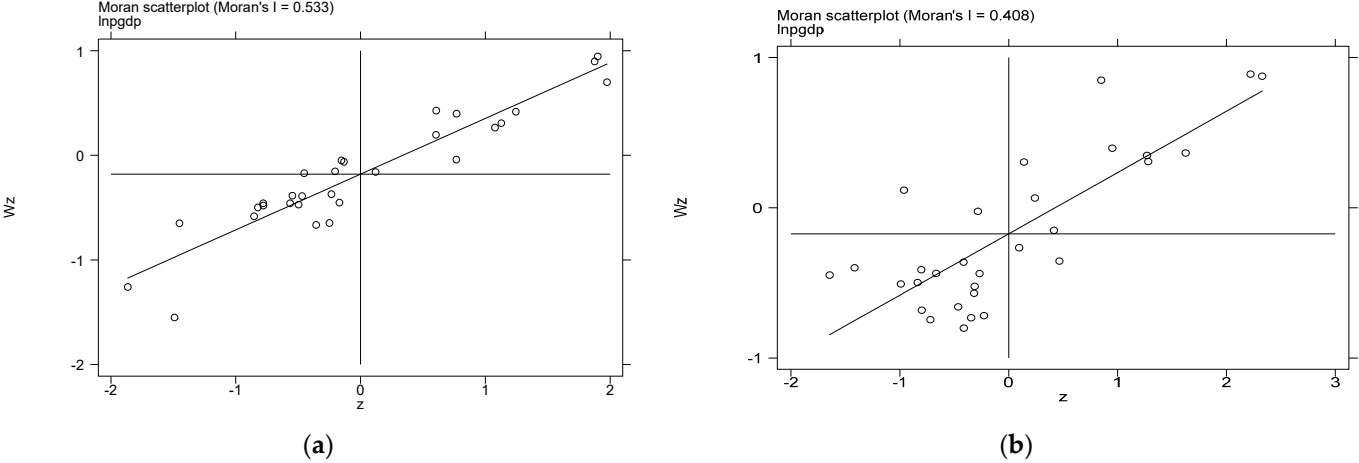

**Figure 2.** Moran scatter plot of regional economic growth in 2011 (**a**) and 2019 (**b**).

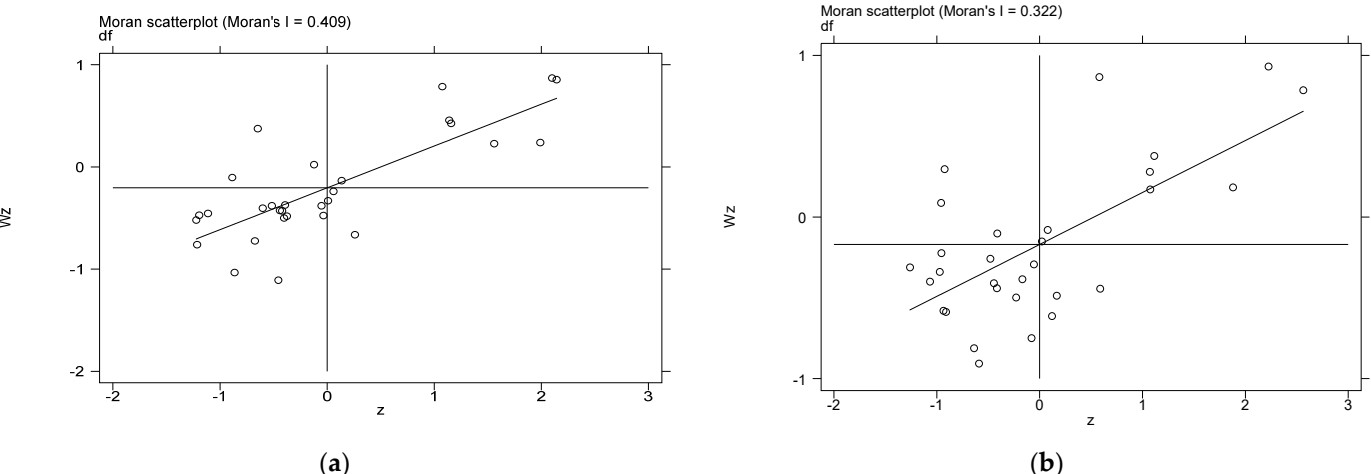

**Figure 3.** Moran scatter plot of digital financial inclusion in 2011 (**a**) and 2019 (**b**).

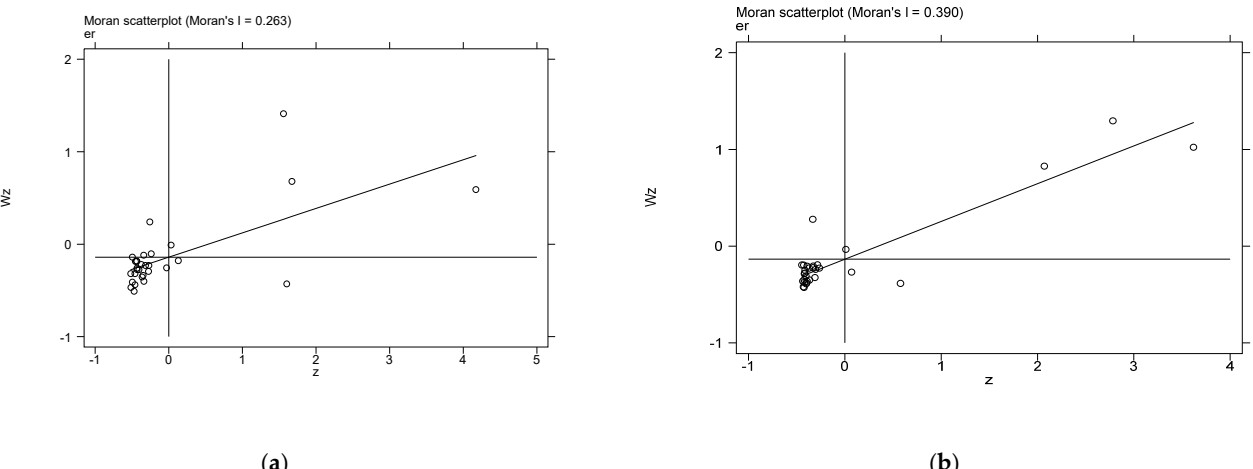

**Figure 4.** Moran scatter plot of environmental regulation in 2011 (**a**) and 2019 (**b**).

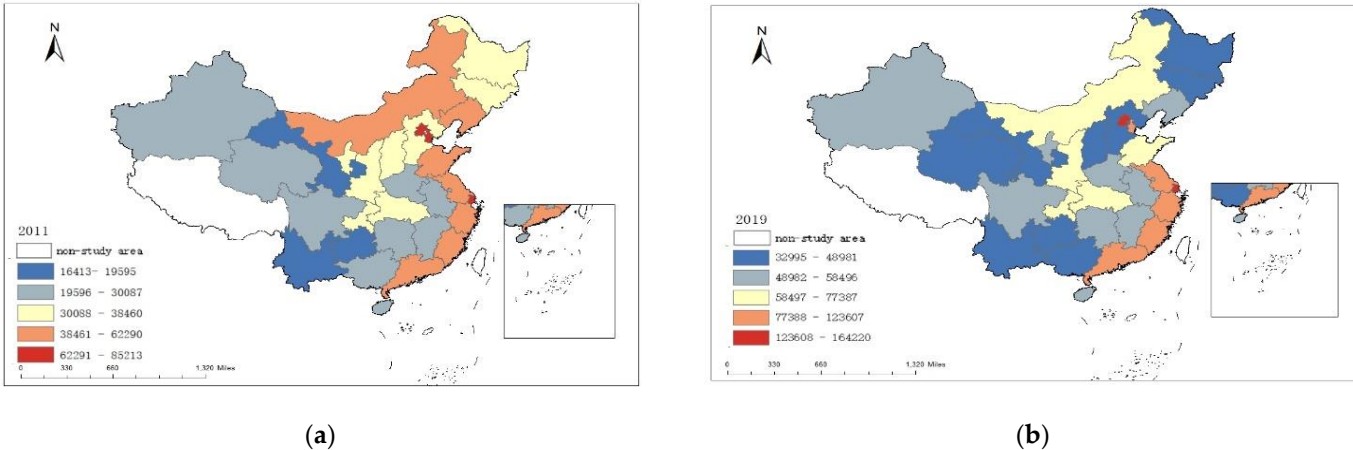

**Figure 5.** Spatial pattern of China's regional economic growth in 2011 (**a**) and 2019 (**b**).

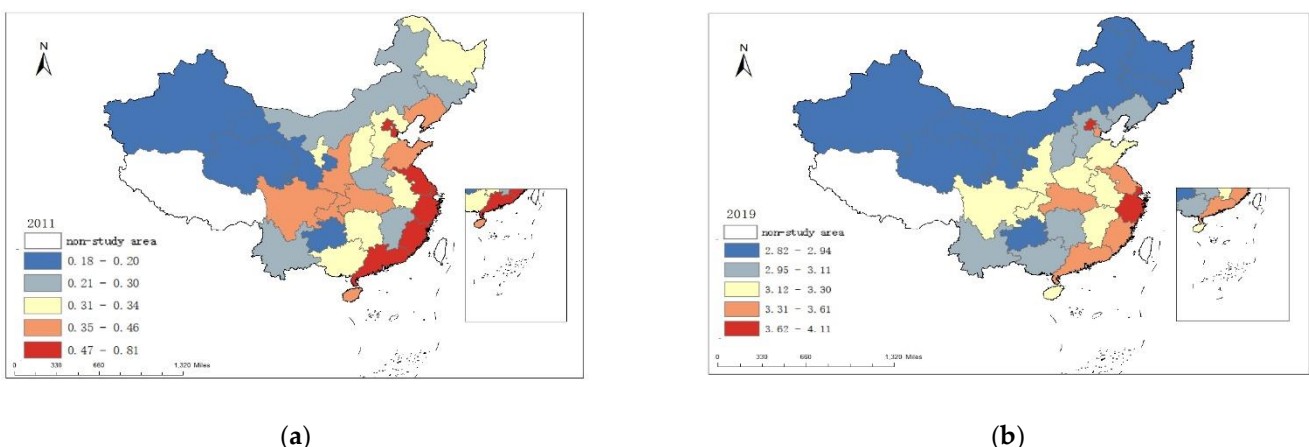

**Figure 6.** Spatial pattern of China's digital financial inclusion in 2011 (**a**) and 2019 (**b**).

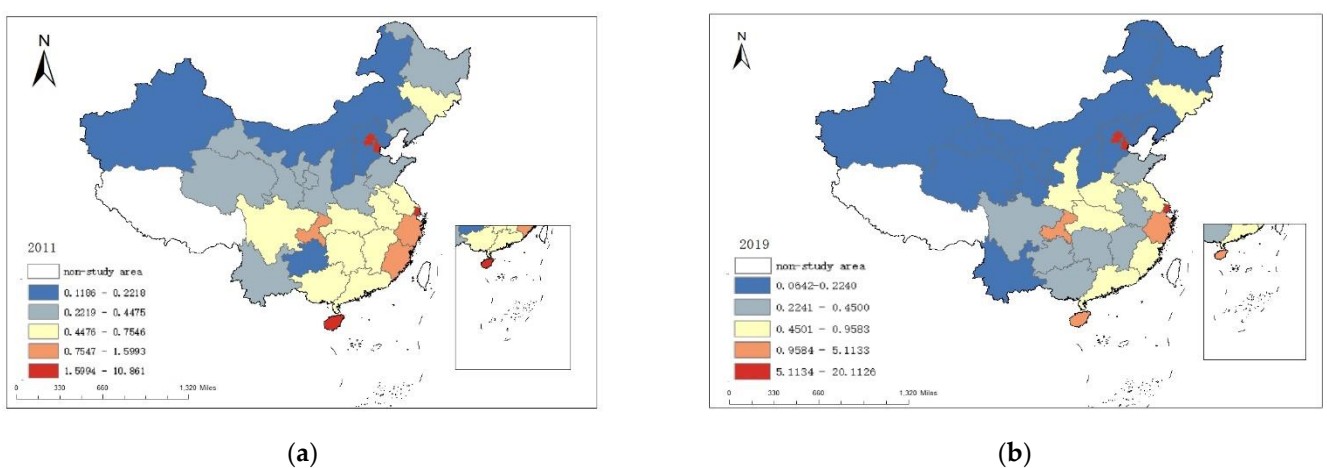

**Figure 7.** Spatial pattern of China's environmental regulation in 2011 (**a**) and 2019 (**b**).

The Moran scatter plots (Figures 2–4) show that there are spatial trends of high–high and low–low agglomeration in regional economic growth and the development level of digital financial inclusion. Some provinces and cities, such as the central region and Northwest China, are in a "low–low" agglomeration regions. On the contrary, the eastern region is mostly located in the "high–high" agglomeration regions. Furthermore, the intensity of environmental regulation in most provinces is found in the "low–low"

agglomeration regions, A few provinces, such as Beijing, Tianjin and Shanghai, are located in a high–high regulated region. As illustrated in Figures 5–7, the overall level of regional economic growth, digital financial inclusion and environmental regulation has improved significantly from 2011 to 2019. The economic growth level of Beijing and Shanghai is still at its highest level. Some provinces and cities, such as Jiangsu, Zhejiang, Fujian and Guangdong, are still in the second tier. Combined with other areas of the country, these form a clear core–periphery model wherein the eastern region's digital financial inclusion index has leading areas, such as Beijing, Shanghai, Zhejiang, Jiangsu, Fujian and Guangdong. In Figure 7, the intensity of environmental regulation in Beijing, Tianjin and Shanghai has always maintained a high level.

### 5.1.2. Analysis of Spatial Econometric Models

This paper used Stata 16.0 software to analyze the impact of digital financial inclusion and environmental regulation on regional economic growth. The three spatial economic models, SAR, SEM, and SDM, were used to improve the authenticity of the empirical results. First, the Hausman test indicated that a random-effects model should be used in this study. In order to study the joint effect of digital financial inclusion and environmental regulation, we introduce cross terms into the three models for regression tests. The results are shown in Table 3.

**Table 3.** Estimation results for different models.

| Variable | SDM | | SEM | | SAR | |
|---|---|---|---|---|---|---|
| | **Model I** | **Model II** | **Model III** | **Model IV** | **Model V** | **Model VI** |
| df | 0.2528 *** | | 0.1892 *** | | 0.1416 *** | |
| | (4.08) | | (16.3) | | (9.35) | |
| er | 0.0059 ** | | 0.0081 *** | | 0.0078 *** | |
| | (2.43) | | (3.61) | | (3.32) | |
| dfxer | | 0.0025 *** | | 0.0028 *** | | 0.0034 *** |
| | | (3.91) | | (4.8) | | (5.36) |
| te | 0.0083 *** | 0.0125 *** | 0.0075 *** | 0.0106 *** | 0.0091 *** | 0.0169 *** |
| | (3.23) | (4.82) | (3.04) | (4.17) | (3.62) | (6.37) |
| is | 1.9405 *** | 2.1621 *** | 1.9373 *** | 2.2957 *** | 1.8374 *** | 2.1603 *** |
| | (4.91) | (5.15) | (5.2) | (6.01) | (4.88) | (5.25) |
| hc | 11.1150 *** | 16.0666 *** | 10.4545 *** | 12.8462 *** | 11.6521 *** | 23.8500 *** |
| | (2.82) | (3.93) | (2.81) | (3.27) | (3) | (5.72) |
| urb | 0.0036 ** | 0.0047 *** | 0.0028 * | 0.0040 ** | 0.0037 ** | 0.0088 *** |
| | (2.32) | (2.84) | (1.83) | (2.34) | (2.34) | (5.1) |
| gov | −2.2475 *** | −2.3418 *** | −2.2535 *** | −2.1068 *** | −2.1364 *** | −1.2151 *** |
| | (−9.54) | (−9.67) | (−10.17) | (−8.17) | (−10.29) | (−6.11) |
| fdi | 2.2492 *** | 2.2901 *** | 2.2848 *** | 1.7578 *** | 2.3153 *** | 1.4190 *** |
| | (4.71) | (4.62) | (4.87) | (3.57) | (4.81) | (2.69) |
| Cons | 5.9973 *** | 4.2601 *** | 8.7807 *** | 8.6751 *** | 6.5410 *** | 1.9448 *** |
| | (5.53) | (4.29) | (25.09) | (22) | (9.45) | (4.29) |
| W * df | −0.1663 ** | | | | | |
| | (−2.55) | | | | | |
| W * er | −0.0144 ** | | | | | |
| | (−2.07) | | | | | |
| W * dfxer | | −00130 | | | | |
| | | (−0.68) | | | | |
| ρ | 0.2622 *** | 0.3807 *** | | | 0.2144 *** | 0.5683 *** |
| | (3.01) | (4.91) | | | (3.41) | (12.79) |
| λ | | | 0.3804 *** | 0.8908 *** | | |
| | | | (4.55) | (38.98) | | |
| R2 | 0.9285 | 0.9196 | 0.9189 | 0.5054 | 0.9176 | 0.8469 |

Note: z statistic in parentheses. ***, **, * indicate statistical significance at 1%, 5%, and 10%, respectively.

Table 3 demonstrates that the direction and significance of the three models are consistent except for the change in coefficient, which verifies the robustness of the estimation results to a certain extent. In Models I, III, and V, the coefficient of digital financial inclusion, environmental regulation and other control variables all pass the significance test. From the perspective of R-squared, the spatial Durbin model fit better, so the following conclusions are drawn from that model.

Model I in Table 3 shows the independent effects of the digital financial inclusion and environmental regulation on regional economic growth. The spatial autocorrelation coefficient ρ is positive at the 1% significance level under the economic distance weight matrix. It shows that the economic development of China's provinces and cities will be affected by the provinces with close economic ties, thus forming a positive spatial spillover effect. From the core explanatory variables, the results show that digital financial inclusion and environmental regulation have a significant positive effect on economic growth. Digital financial inclusion can break the spatial constraints of financial services. It combines digital technology with Internet technology to reduce the threshold of financial services and the financing cost of enterprises, which can ultimately increase the activity of enterprises and promote economic growth. However, the spatial spillover coefficient of digital financial inclusion is significantly negative. It shows that digital financial inclusion inhibits the economic development of regions with similar economic levels. The possible reason for this is that when the development level of digital inclusive finance in the region is relatively high, it will attract the capital resources of regions with similar economic levels and weaken the foundation of its development, which is not conducive to the development of digital inclusive finance in the region with similar economic levels and has a negative impact on its economic growth. Therefore, Hypothesis 1 is proven correct. The conclusions of this study are similar to those of Yang et al. and Chu et al. [14,15]. Secondly, the regression coefficient of environmental regulation is significantly positive. It shows that environmental regulation can effectively curb the short-sighted behavior of enterprises. At the same time, it can improve the efficiency of resource allocation, promote the technological innovation of enterprises, and force the upgrading of industrial structure to promote the economic growth of the region. Similar to digital financial inclusion, the spatial spillover coefficient of environmental regulation is also significantly negative. It shows that the strengthening of environmental regulation in this region will inhibit the economic growth of regions with similar economic levels. When the environmental regulation in the region is strengthened, the pollution-intensive industries will be transferred to the areas with similar economic levels and a lower environmental regulation, which will have a negative impact on the economic growth of the area with similar economic levels. Therefore, Hypothesis 2 is proven to be correct. This is similar to the findings of Shangguan et al. [41].

Due to the interactions between the two variables, "digital financial inclusion plus environmental regulation", a cross item of digital financial inclusion and environmental regulation, was introduced into the model to study its synergic effects on economic growth. The results are presented in Models II in Table 3. The results show that the regression coefficient of the interaction term between digital financial inclusion and environmental regulation is significantly positive at the 1% significance level. It is shown that the current combined effect of digital financial inclusion and environmental regulation has a synergistic effect. Therefore, Hypothesis 3 is proved. This conclusion is similar to that of Li et al. and Cao et al. [42,43]. However, compared with Model I, the combined effect of the two is smaller than the independent effect. This shows that the promotion of digital financial inclusion in regional economic growth is reduced after being affected by environmental regulations. When funds are limited, the strengthening of environmental regulations will force enterprises to increase investment in environmental protection and adjust their capital allocation plans, which can weaken the impact of the financial system on economic growth. Similarly, under the influence of digital financial inclusion, the effect of environmental regulation on economic growth will also be reduced. It may be that the intensity of environmental regulation in this region does not match the level of finance. In other

words, the environmental protection costs increased by environmental regulation cannot be supplemented by the financial system. Therefore, the impact of environmental regulation on regional economic growth is weakened.

For the control variables, the coefficients of technological innovation and industrial structure in Model I are significantly positive at the 1% significance level. This shows that in the adjustment of industrial structure, new technologies and new business forms are bound to emerge, which can promote the economic growth of various regions. Cheng et al. also came to a similar conclusion [63]. The regression coefficient of human capital and urbanization level is significantly positive, and the regression coefficient of human capital is larger. This shows that the improvement of human capital can significantly promote economic growth and is similar to the conclusion of Li et al. [64]. Moreover, during the process of urbanization, the population agglomeration was promoted and a sufficient labor force was generated for economic growth. In addition, the improvement of human capital knowledge and skills can further promote economic growth. This conclusion is similar to that of Yang et al. and Chen [11,65]. Government intervention had a negative effect on regional economic growth, and Yang et al. also came to a similar conclusion [14]. The reason for this is that the Government's human intervention may result in an inefficient allocation of social resources and negatively affect the overall economy. However, foreign direct investment can promote regional economic growth by bringing in production factors, such as capital and technology.

### 5.1.3. Spatial Effect Decomposition

To further analyze the spatial impact of digital financial inclusion and environmental regulation on regional economic growth, this paper breaks down the spatial effect with the help of a partial differential equation. Table 4 presents the independent models for digital financial inclusion and environmental regulation, along with the direct effect, indirect effect, and total effect of each variable on regional economic growth in a collaborative model between the two variables.

**Table 4.** Decomposition of spatial effects.

| Variable | Model I | | | Model II | | |
|---|---|---|---|---|---|---|
| | **Direct Effect** | **Indirect Effect** | **Total Effect** | **Direct Effect** | **Indirect Effect** | **Total Effect** |
| df | 0.2497 *** (4.09) | −0.1312 ** (−1.96) | 0.1185 *** (4.72) | | | |
| er | 0.0051 ** (2) | −0.0172 * (−1.85) | −0.0122 (−1.14) | | | |
| dfxer | | | | 0.0025 *** (3.33) | −0.0004 (−0.14) | 0.0021 (0.58) |
| te | 0.0096 *** (3.74) | 0.0261 *** (2.69) | 0.0358 *** (3.31) | 0.0144 *** (5.32) | 0.0355 *** (2.83) | 0.0499 *** (3.56) |
| is | 1.9264 *** (4.9) | −0.1838 (−0.22) | 1.7426 * (1.72) | 2.1761 *** (5.1) | −0.5458 (−0.51) | 1.6304 (1.23) |
| hc | 11.6836 *** (2.87) | 11.1298 (0.96) | 22.8133 * (1.69) | 17.9420 *** (4.42) | 32.6641 ** (2.29) | 50.6060 *** (3.19) |
| urb | 0.0040 ** (2.55) | 0.0086 (1.18) | 0.0126 (1.62) | 0.0059 *** (3.49) | 0.0218 *** (2.59) | 0.0277 *** (3.03) |
| gov | −2.2372 *** (−9.26) | 0.0671 (0.14) | −2.1701 *** (−4.4) | −2.2679 *** (−9.44) | 0.9769 * (1.8) | −1.2910 ** (−2.18) |
| fdi | 2.2274 *** (4.62) | 0.1919 (0.1) | 2.4193 (1.16) | 2.4499 *** (4.45) | 2.9521 (1.3) | 5.4021 ** (2.1) |

Note: z statistic in parentheses. ***, **, * indicate statistical significance at 1%, 5%, and 10%, respectively.

In the independent model, the total effect and the direct effect of digital financial inclusion were both significantly positive, but the indirect effect was significantly negative, indicating that digital financial inclusion has a significant spatial spillover effect. This shows that the development of digital inclusive finance will promote the economic growth

of the region, but it will inhibit the economic development of regions with similar economic levels. The possible reason for this is that, with the improvement in the level of digital inclusive finance in the region, there will be a strong demand for key elements such as talent and capital, which may produce a "siphon effect" and cause the cross-regional flow of key elements. Consequently, economic development in regions with similar economic levels is inhibited. The direct effect of environmental regulation is significantly positive, and the indirect effect is significantly negative, but the total effect is not significant. These results show that environmental regulation plays a significant role in promoting economic growth in the region. However, the increased intensity of environmental regulation in this region will inhibit the economic development of regions with similar economic levels. The possible reason for this is that, according to the "pollution paradise hypothesis", the strict implementation of environmental regulations in the region will lead to the transfer of polluting industries to regions with similar economic levels and lower environmental regulation, which is not conducive to the transformation and upgrading of industrial structures in regions with similar economies. Therefore, the economic development of economically similar regions may be negatively affected. Under the synergetic model, the direct effect of the cross term of digital financial inclusion and environmental regulation is significantly positive, but the indirect effect and the total effect fail the significance level test, indicating that the cross term of digital financial inclusion and environmental regulation has no significant spatial spillover effect. This may be due to the location priority and competitiveness of digital financial inclusion development and environmental regulation, so that the integrated benefits of digital financial inclusion development and environmental regulation cannot be extended to provinces with similar economic levels, resulting in spillover effects.

### 5.1.4. Research on Regional Space

The study analyzed regional differences across the three eastern, central, and western regions. A spatial Durbin model was then used to empirically analyze the spillover effect of each variable. The results are listed in Table 5.

**Table 5.** Regional spatial econometric models.

| Variable | East | | Central | | West | |
|---|---|---|---|---|---|---|
| | **Model VII** | **Model VIII** | **Model IX** | **Model X** | **Model XI** | **Model XII** |
| df | 0.3770 *** | | 0.1295 | | 0.2645 *** | |
| | (4.87) | | (0.71) | | (2.72) | |
| er | 0.0050 ** | | 0.1834 *** | | 0.0371 ** | |
| | (1.97) | | (2.98) | | (1.86) | |
| dfXer | | 0.0013** | | 0.0480** | | 0.0200 *** |
| | | (2.01) | | (2.11) | | (2.88) |
| te | 0.0126 *** | 0.0179 *** | 0.0006 | −0.0157 | 0.0141 | 0.0194 ** |
| | (5.02) | (7.36) | (0.05) | (−1.53) | (1.63) | (2.18) |
| is | 0.0304 | 3.0089 *** | 0.3153 | 1.9714 * | 1.6571 *** | 1.6718 *** |
| | (0.06) | (4.31) | (0.88) | (1.85) | (2.95) | (2.76) |
| hc | 15.6458 *** | 16.4493 ** | −4.9819 | −24.0641 * | 32.6429 *** | 37.8767 *** |
| | (3.69) | (2.52) | (−0.53) | (−1.95) | (5.02) | (5.44) |
| urb | 0.0105 *** | −0.0037 | 0.0272 *** | 0.0338 *** | 0.0028 ** | 0.0026 * |
| | (3.36) | (−1.05) | (4.23) | (3.43) | (2.11) | (1.87) |
| gov | −2.1355 *** | −2.6226 *** | −3.2222 *** | −2.0096 *** | −1.9209 *** | −2.1169 *** |
| | (−6.31) | (−7.87) | (−7.52) | (−3.09) | (−6.66) | (−7.26) |
| fdi | 1.6784 ** | 0.9285 * | 2.5224 | −0.8252 | 2.8550 ** | 3.2122 ** |
| | (2.32) | (1.70) | (1.33) | (−0.37) | (2.09) | (2.04) |
| Cons | 7.5378 *** | 3.2897 *** | 8.8373 *** | 6.6526 *** | 11.1541 *** | 6.6670 *** |
| | (5) | (5.27) | (5.58) | (3.86) | (5.91) | (4.37) |
| ρ | 0.2943 *** | 0.4008 *** | 0.0508 | 0.0351 | −0.0954 | 0.2213 * |
| | (2.69) | (4.31) | (0.35) | (0.27) | (−0.58) | (1.68) |
| R2 | 0.9244 | 0.9549 | 0.8926 | 0.9339 | 0.9655 | 0.9574 |

Note: z statistic in parentheses. ***, **, * indicate statistical significance at 1%, 5%, and 10%, respectively.

The independent effect test results of models VII, IX and XI show that the spatial lag coefficient of the eastern region is 0.2943, which is significant at the 1% significance level. However, the spatial lag coefficients in the central and western regions did not pass the significance test. It shows that there is a significant positive spatial spillover effect in the economic development of the eastern region. From the core explanatory variables, digital financial inclusion can significantly promote the economic development of the eastern and western regions. The promotion effect in the eastern region is larger, which is similar to the conclusion of Yang et al. [17]. The effect of digital financial inclusion on the central region is not significant. The possible reason for this is that the industrial organizations and financial institutions in the eastern region are relatively complete, and the financial supply essentially meets these requirements, which can effectively promote economic growth. Digital inclusive finance in the western regions is in the early stage of construction, and the digital infrastructure is not yet perfect. Therefore, the promotion effect is smaller than in the eastern region. Although the central region is on the rise by undertaking an industrial transfer from the eastern region, it may be over-reliant on resources, incomplete industrial chains, and low value-added products, resulting in the unsatisfactory synergy efficiency of various resources. Therefore, digital financial inclusion will not be able to play its full role.

The regression coefficients of environmental regulation in the eastern, central and western regions are 0.0050, 0.1834 and 0.0371, respectively, all of which have passed the significance test. It can be seen that environmental regulation in the eastern, western and central regions can significantly promote economic development. Moreover, its promoting effect on the economy is highest in the central region, followed by the western region, and then the eastern region. The possible reason for this is that there are many resource-based cities in the central region, and the secondary industry accounts for a large proportion of production. With the strengthening of environmental regulations, high-polluting enterprises will be promoted to carry out technological innovation, which can achieve industrial structure upgrading and economic growth. In comparison, the western region has a greater demand for economic development and may have looser environmental regulations. Therefore, it may become a place for the westward migration of pollution-intensive enterprises in the central and eastern regions. The increase in industry is likely to boost economic growth. The eastern region has a higher economic level and may have a higher "tolerance level" for environmental regulation. Therefore, the promotion effect on economic development is small.

The combined effect results of models VIII, X and XII show that the spatial lag coefficient of the eastern region is 0.4008, which is significant at the 1% significance level. The spatial lag coefficient of the western region is significantly positive at the 10% significance level. This shows that, when the intersection of digital financial inclusion and environmental regulation is introduced, the eastern and western regions show a significant positive spatial spillover effect. The coefficients of the cross terms in the eastern, central and western regions are 0.0013, 0.0480 and 0.0200 respectively, all of which passed the significance test. This shows that the synergistic effect of digital financial inclusion and environmental regulation can significantly promote economic growth in the eastern, central and western regions. As a result, its synergy weakened in the central, western and eastern regions. The possible reason for this is that the infrastructure in the eastern region is complete, the development level of digital inclusive finance is relatively high, and the advantages and spillover effects also emerged. Therefore, the synergy between digital financial inclusion and regulations may be small. However, the central region actively undertakes the transfer of domestic and foreign industries and plays a role in connecting the eastern and the western regions. Its level of digital inclusive finance is higher than that of the western region. With the strengthening of environmental regulations in the central region, on the one hand, digital inclusive finance can effectively relieve the financial pressure of enterprises and promote technological innovation. On the other hand, environmental regulation can guide the precise placement of green credit products in high-tech industries

and promote industrial transformation. Therefore, the synergistic effect of digital financial inclusion and environmental regulation in the central region is relatively large.

*5.2. Empirical Analysis of the Panel Threshold Mode*

5.2.1. Analysis of Threshold Regression Results

To further develop the mechanism of synergy between digital financial inclusion and environmental regulation, the panel threshold mode is adopted. This establishes whether digital financial inclusion and environmental regulation have a threshold effect on regional economic growth. The threshold effect is tested with digital financial inclusion and environmental regulation as the threshold variables. The results are given in Table 6.

**Table 6.** Threshold effect test.

| Variable | Digital Financial Inclusion as the Threshold Variable | | | Environmental Regulation as the Threshold Variable | | |
|---|---|---|---|---|---|---|
| | Single Threshold | Double Threshold | Triple Threshold | Single Threshold | Double Threshold | Triple Threshold |
| 95% confidence interval of single threshold estimation | 2.7638 (2.3892, 2.7691) | 2.3536 (2.1771, 2.3753) | | 0.8422 (0.7436, 0.8629) | 0.8422 (0.7149, 0.8629) | |
| 95% confidence interval of double threshold estimation | | 0.3358 (0.3341, 0.3389) | | | 0.1518 (0.1498, 0.1552) | |
| F-statistic | 22.22 | 16.82 | 11.85 | 12.89 | 7.25 | 8.24 |
| *p*-value | 0.0467 | 0.0133 | 0.5767 | 0.3067 | 0.58 | 0.73 |
| 10% critical value | 19.3783 | 10.3908 | 23.4653 | 17.8197 | 18.1907 | 20.9257 |
| 5% critical value | 21.4364 | 12.1664 | 27.1831 | 21.2592 | 22.3662 | 26.1239 |
| 1% critical value | 28.0512 | 17.8374 | 33.8956 | 28.7577 | 30.4831 | 36.1273 |

The results show that when digital financial inclusion is the threshold variable, the environmental regulation has a double threshold effect, while the threshold values are 0.3358 and 2.3536. When environmental regulation is the threshold variable, digital financial inclusion has no threshold effects. Therefore, Hypothesis 4b is proved and Hypothesis 4a is rejected. Using the results of the panel threshold test, a regression analysis was carried out on the model, and the results are shown in Table 7.

**Table 7.** Regression estimation results of the digital financial inclusion as the threshold variable.

| Variable | Value | t-Value |
|---|---|---|
| te | 0.0300 *** | 9.2 |
| is | 3.5189 *** | 6.44 |
| hc | 55.2600 *** | 11.39 |
| urb | 0.0117 *** | 5.22 |
| gov | −1.0984 *** | −4.1 |
| fdi | 0.0652 | 0.09 |
| Cons | 6.0560 *** | 13.01 |
| er(df ≤ 0.3358) | −0.2472 *** | −3.24 |
| er(0.3358 < df ≤ 2.3536) | −0.0085 | −1.14 |
| er(2.3536 < df) | 0.0122 *** | 3.15 |
| R2 | 0.8449 | |

Note: *** indicate statistical significance at 1%.

The threshold regression results show that when the development level of digital inclusive finance is lower than 0.3358, the regression coefficient of the impact of environmental regulation on regional economic development is significantly negative, and the

value is −1.0984. When the development level of digital inclusive finance was raised to 0.3358~2.3536, the regression coefficient changed from negative to positive and failed the significance test. However, when the development level of digital financial inclusion was higher than 2.3536, the regression coefficient of environmental regulation on economic development was significantly positive with a value of 0.0122. This indicated that the impact of environmental regulation on economic development was regulated by the threshold effect of the development level of digital financial inclusion. Environmental regulation will have a negative effect on economic development when the development level of digital financial inclusion is low. With the development of digital financial inclusion, environmental regulation has gradually played an active role in economic development. However, when the level of digital financial inclusion is in the middle of the two thresholds, environmental regulation cannot have a significant impact on regional economic growth. The possible reason for this is that digital inclusive finance is constantly choosing between providing environmental protection funds or promoting enterprise production, which cannot make up for environmental protection costs in a timely manner.

5.2.2. Analysis of Thresholds by Region

In order to further explore whether there are significant regional differences in the threshold effect, the regional threshold model test and regression results are shown in Table 8.

The regional threshold regression results show that when digital financial inclusion is used as the threshold variable, the digital financial inclusion variables in the eastern and western regions pass the single-threshold test, but the threshold effect in the central region is not significant. In addition, environmental regulation variables still fail to pass the threshold effect test. This shows that the current impact of environmental regulation on economic growth in the eastern and western regions will be affected by the level of development of digital financial inclusion. The central region may be in an ambiguous period of coordination between the level of digital financial inclusion and environmental regulations, and the impact of the level of digital financial inclusion is unknown. At the same time, this also shows that the development of digital financial inclusion in the emerging stage has not been affected by the level of environmental regulation.

Specifically, in the eastern region, when the level of digital financial inclusion is lower than 2.7638, environmental regulation is significant at the 1% significance level and the regression coefficient is 0.0112, indicating that environmental regulation can significantly promote economic growth within this range. When digital financial inclusion exceeds this threshold variable, the regression coefficient of environmental regulation is 0.0220, which enhances the promotion of economic development and exhibits a significant positive marginal incremental effect. However, in the western region, the threshold for digital financial inclusion is 0.2889. When it is less than this threshold, the regression coefficient of environmental regulation is −0.5541, which has a negative impact on economic growth. When this threshold is crossed, the regression coefficient of environmental regulation changes from negative to positive with a value of 0.0898, which has a positive effect on economic growth. The results show that the eastern region can effectively supplement the environmental protection costs brought about by environmental regulation with its relatively high level of digital finance. However, the cost of digital inclusive finance in the western region is relatively high in the early stage of construction. Therefore, it is difficult to supplement the environmental protection costs caused by environmental regulations that have a negative impact on the technological innovation of enterprises, ultimately reducing the profitability of enterprises and affects economic growth. Conversely, when the level of digital financial inclusion crosses the threshold, it can effectively promote economic growth.

**Table 8.** Threshold effect test and regression analysis of regions.

| Variable | East | | | | Central | | | | West | | | |
|---|---|---|---|---|---|---|---|---|---|---|---|---|
| | Digital Financial Inclusion as the Threshold Variable | | Environmental Regulation as the Threshold Variable | | Digital Financial Inclusion as the Threshold Variable | | Environmental Regulation as the Threshold Variable | | Digital Financial Inclusion as the Threshold Variable | | Environmental Regulation as the Threshold Variable | |
| | Value | t-Value | Value | t-Value | Value | t-Value | Value | t-Value | Value | t-Value | Value | t-Value |
| te | 0.0292 *** | 8.38 | 0.0144 *** | 4.14 | −0.0139 | −1.47 | −0.0084 | −0.96 | 0.03452 *** | 3.69 | 0.0036 | 0.52 |
| is | 6.8975 *** | 5.47 | 4.9418 *** | 4.65 | −0.0371 | −0.03 | 3.9847*** | 3.19 | 2.8851 *** | 3.84 | 1.0409 * | 1.84 |
| hc | 26.2093 ** | 2.31 | 3.5962 | 0.4 | −12.0743 | −0.89 | −18.2380 | −1.53 | 67.1613 *** | 14.04 | 23.4818 *** | 5.02 |
| urb | 0.0144 ** | 2.57 | −0.0016 *** | −0.32 | 0.0472 *** | 4.58 | 0.0143 | 1.11 | 0.0061 *** | 3.45 | 0.0042 ** | 2.58 |
| gov | −1.4248 *** | −2.83 | −2.2528 | −5.19 | −2.2450 *** | −2.96 | −1.2179 ** | −1.94 | −0.7668 ** | −2.66 | −1.8278 *** | −7.76 |
| fdi | 1.1452 | 1.58 | 2.1431 *** | 3.61 | −3.2210 | −1.39 | −3.6244 | −1.66 | 3.6675 * | 1.8 | 0.8623 | 0.72 |
| Cons | 3.2185 *** | 3.31 | 6.4992 | 6.88 | 9.0350 *** | 8.73 | 6.7310 | 7.17 | 6.7299 *** | 10.5 | 9.3141 *** | 18.6 |
| er(df ≤ 2.7638) | 0.0112 *** | 2.90 | | | | | | | | | | |
| er(2.7638 < df) | 0.0220 *** | 6.56 | | | | | | | | | | |
| er(df ≤ 0.2889) | | | | | | | | | −0.5541 *** | −4.55 | | |
| er(0.2889 < df) | | | | | | | | | 0.0898 *** | 3.06 | | |
| Single threshold (F/P) | 20.73 ** | 0.0533 | 15.80 | 0.1233 | 13.51 | 0.2067 | 8.24 | 0.33 | 35.36 *** | 0 | 15.61 | 0.19 |
| Double threshold (F/P) | 7.87 | 0.2167 | 10.91 | 0.1500 | 7.8 | 0.3933 | 4.28 | 0.62 | 11.45 | 0.2 | 7.43 | 0.4633 |
| Triple threshold | 4.50 | 0.5600 | 4.62 | 0.9067 | 12.83 | 0.5367 | 3.65 | 0.8033 | 4.32 | 0.7867 | 4.95 | 0.66 |
| Number of thresholds (F/P) | single threshold | | no threshold | | no threshold | | no threshold | | single threshold | | no threshold | |
| Value of the threshold | 2.7638 | | | | | | | | 0.2889 | | | |
| confidence interval | (2.5659,2.7811) | | | | | | | | (0.2263,0.3131) | | | |
| R2 | 0.8764 | | 0.9261 | | 0.9270 | | 0.9409 | | 0.9246 | | 0.9645 | |

Note: ***, **, * indicate statistical significance at 1%, 5%, and 10%, respectively.

In addition, when the level of digital financial inclusion in the western region crosses the threshold, environmental regulation plays a greater role in promoting economic growth than in the eastern region. The possible reason for this is that the level of digital inclusive finance in the eastern region is relatively high, due to the natural profit-seeking nature of finance, it will prefer the virtual economy sector, which will increase the gap between the development of the real economy and the virtual economy. To a certain extent, it will hinder the transformation of green innovation technological achievements forced by environmental regulations and impact the development of the real economy. Therefore, compared with the western regions, the promoting effect of environmental regulation in the eastern regions will be lower.

## 6. Conclusions and Suggestions

### 6.1. Main Conclusions

This paper constructs environmental regulation indicators from the perspective of waste water, waste gas and waste solids. It calculates the comprehensive index of environmental regulation in each province using the entropy weight method. Based on the theory and literature analysis of the impact of digital financial inclusion and environmental regulation on economic growth, this study used provincial panel data from 2011 to 2019 to conduct an empirical analysis. First, the independent and joint effects of digital financial inclusion and environmental regulation on economic growth were analyzed using spatial SDM, SAR and SEM models. Secondly, using the panel threshold model, the study discussed the impact of environmental regulation on economic growth under different levels of digital financial inclusion. Finally, the study divided the research samples into three regions (eastern, central, and western regions) to study the regional heterogeneity impact of digital financial inclusion and environmental regulation on economic growth. The empirical results show the following:

On the whole, digital financial inclusion and environmental regulation can significantly promote the economic growth of the region by using an independent effects test. However, they all have a negative impact on the economic development of regions with similar economic levels. This reflects the negative spatial spillover effect. Therefore, Hypothesis 1 and Hypothesis 2 are confirmed. The research conclusion of Hypothesis 1 is similar to that of Yang et al. and Chu et al. [14,15]. The result of Hypothesis 2 is similar to that of Shangguan et al. [41]. In the joint effect test, it is found that the cross term of digital financial inclusion and environmental regulation can significantly promote regional economic growth. This shows that digital financial inclusion and environmental regulation have a synergistic effect on economic development, and Hypothesis 3 is confirmed. This conclusion is similar to that of Li et al. and Cao et al. [42,43]. Furthermore, using digital financial inclusion as a threshold variable, the impact of environmental regulation on regional economic growth is discussed. The results show that there is a double-threshold effect in digital financial inclusion, and the impact of environmental regulation on regional economic growth has changed from negative to positive, showing a significant promoting effect. It shows that the effect of environmental regulation on regional economic development will be affected by the level of digital financial inclusion, which confirms Hypothesis 4b. However, when environmental regulation was used as the threshold variable, the significance test was not passed. This showed that the effect of digital financial inclusion on regional economic growth was not affected by the intensity of environmental regulation at this stage, and Hypothesis 4a was rejected.

In the independent effect test, the spatial lag coefficient of the eastern region passed the test of significance. The spatial lag coefficients in the central and western regions did not pass the significance test. This shows that there is a significant positive spatial spillover effect in the economic development of the eastern region. From the core explanatory variables, digital financial inclusion can significantly promote economic growth in the eastern and western regions. The promotion effect of the eastern regions is excellent. However, the promotion effect of the central regions did not pass the significance test. Environmental

regulations in the eastern, central and western regions all passed the significance test. This shows that these regulations can significantly promote the economic growth of the region. Moreover, the promoting effect on the economy was highest in the central region, followed by the western region, and then the eastern region. In the joint effect test, the spatial lag coefficients of the eastern and western regions passed the significance test. This showed that when digital financial inclusion and environmental regulation synergized, there was a significant positive spatial spillover effect in the economic development of the eastern and western regions. The cross term of digital financial inclusion and environmental regulation passed the significance test in the three regions. This showed that its synergistic effect could significantly promote the economic growth of the eastern, central and western regions. The effect of synergy decreased in the central, western and eastern regions as a result. Similarly, the impact of environmental regulation on economic growth in the eastern and western regions was affected by the level of digital financial inclusion after the threshold effect test. Environmental regulation in the eastern region could always promote economic growth when digital financial inclusion crossed the threshold, whereas the impact of environmental regulation in the western region on economic growth was first inhibited and then promoted. However, the promotion effect of the western region was found to be greater than that of the eastern region, whereas the regression coefficient of the central region was not found to be significant.

### 6.2. Policy Recommendations

Policy recommendations for the development of digital financial inclusion are as follows. First, the Government should improve the top-level design of the digital inclusive financial system, establish an effective organizational management mechanism, and improve the ability and efficiency of digital inclusive finance to serve the real economy. In addition, they should rationally deploy digital technologies and strengthen regional synergy to promote the positive impact of direct effects on economic growth and reduce the negative impact of spatial spillover effects. Second, there is regional heterogeneity regarding the impact of digital financial inclusion on economic growth. For underdeveloped areas in the central and western regions, the Government should increase financial support and infrastructure construction to encourage financial institutions to promote digital financial services and reduce transaction costs. For the developed eastern regions, financial institutions should carry out financial innovation services to promote regional innovation and economic growth. Finally, financial regulators should make full use of digital technology and learn from advanced experience to speed up the formulation of regulatory policies for the digital inclusive financial industry, which can improve the flexibility, innovation and efficiency of supervision.

In terms of environmental regulation, first, the Government should increase investments in research and development, optimize the allocation of research and development resources, and support transformation enterprises to introduce advanced production technologies, which can reduce pollutant emissions from the source and their negative impact on economic development. Second, for economically developed regions, a higher intensity of environmental regulation should be adopted, and "incentive" environmental regulation tools should be used as the mainstay. For economically underdeveloped areas, moderate environmental regulation intensity should be adopted, supplemented by "incentive" and "control" environmental regulation tools. However, in order to protect the ecological environment in ecologically fragile areas, "controlling" environmental regulation tools should be the mainstay, and the Government's regulatory role should be brought into full play. Finally, in the long run, it is necessary to steadily increase the intensity of regional environmental supervision and give full play to the role of environmental supervision. When formulating environmental policies, local governments should fully consider the decisions of other regional governments to achieve a healthy competition between regions and reduce the negative spatial spillover effect of environmental regulation.

There are two main suggestions regarding the synergy between digital financial inclusion and environmental regulation. On the one hand, all regions should strengthen regional cooperation and jointly explore and improve regional cooperation and coordination mechanisms for environmental regulation and digital financial inclusion. It is necessary to carry out cross-regional environmental protection law enforcement and financial instruments implementation cooperation, which will help achieve the coordinated development of the environment, finance and economy in each region. On the other hand, the government should formulate differentiated environmental regulation policies based on the differences in the development levels of different digital financial inclusion policies. In the eastern region, the level of digital financial inclusion is high, technological innovation is strong, and the market is mature. Therefore, it is suitable to implement market-incentivized environmental regulation policies, such as emission rights trading and emission fees, which is conducive to stimulating the positive externality effect of environmental regulation. In the central and western regions, the development of digital financial inclusion is relatively slow. The areas where pollution-intensive enterprises are located should focus on adopting command-and-control environmental regulation policies. The environmental regulation cost of such enterprises is greater than the compensation effect of technological innovation. Therefore, it is necessary to provide support to enterprises by means of external coercive measures from the government and increase financial payment.

### 6.3. Research Limitations and Future Research

This research was subject to some limitations, which should be considered in further research. First, the article only studies the overall perspective of digital finance and environmental regulation on regional economic growth. However, the breadth of coverage, depth of use and degree of digitalization of digital financial inclusion are not included in this research framework; therefore, the synergy between these three dimensions and environmental regulation requires further research. Second, combined with theoretical analysis, it can be seen that the development of digital inclusive finance and the improvement of environmental regulation can promote the technological innovation of enterprises. Whether technological innovation plays an intermediary role in the process of digital finance and environmental regulation, which affects economic growth, requires further research. Third, the empirical results show that environmental regulation will be affected by digital financial inclusion in the process of affecting regional economic growth. The intensity of environmental regulation can be divided into three areas: strong, medium and weak. It can be further explored whether the threshold value of digital financial inclusion will change. Furthermore, whether the research of this paper can be extended to the economic growth in the international scope remains to be studied.

**Author Contributions:** Conceptualization, R.D.; methodology, F.S.; software, F.S.; validation, S.H. and F.S.; formal analysis, S.H.; investigation, F.S.; resources, F.S.; data curation, F.S.; writing—original draft preparation, F.S.; writing—review and editing, R.D.; visualization, F.S.; supervision, S.H.; project administration, R.D.; funding acquisition, R.D. All authors have read and agreed to the published version of the manuscript.

**Funding:** This work was supported by the China National Key R&D Program during the 13th Five-Year Plan Period (2017YFC0805600).

**Institutional Review Board Statement:** Not applicable.

**Informed Consent Statement:** Informed consent was obtained from all subjects involved in the study.

**Data Availability Statement:** The data and estimations commands that support the findings of this paper are available on request from the first and corresponding authors.

**Conflicts of Interest:** The authors declare no conflict of interest.

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
