# Peer review of "Digital Inclusive Finance, Environmental Regulation, and Regional Economic Growth: An Empirical Study Based on Spatial Spillover Effect and Panel Threshold Effect"

_sustainability, doi:10.3390/su14074340_

Round 1

Reviewer 1 Report

Apparently, the analysis targeted novel contexts, so in the
conclusions, the essence of novel contexts could be better
emphasized in a shorter sentence. Thus, the value creation of
the analysis is also more emphasized.

Reviewer 2 Report

Dear Authors,

I have some recommendations on the article entitled .,Impact of Digital Inclusive Finance and Environmental Regulation on Regional Economic Growth in China in Terms of Spatial Spillover and the Threshold Effect”:

In the Abstract, it is necessary to specify exactly the methodology used, the period and the country/countries/regions analysed.

Avoid numbering in the Abstract.

The Abstract is very general. The title of the paper shows that this is a China-oriented one. This does not follow from the Abstract.

Also, in the Abstract, it is necessary to mention the usefulness and novelty of the research.

Are you sure that the statement in lines 132-133 is true? With so many studies currently being published around the world, there is a possibility that someone somewhere has adopted the topic you are discussing.

Check that all quotes are correct in the text.

You have text inserts written differently than the rest of the article.

From my point of view, the methodological part should start with the presentation of the data.

In Conclusions, you should specify whether you have validated the research hypotheses, make a connection between your conclusions and those of the existing literature, the limits of the research, the domestic and international utility of the research.

I recommend that you also consult recent articles in the international literature.

Reviewer 3 Report

Impact of Digital Inclusive Finance and Environmental Regulation on Regional Economic Growth in China in Terms of Spatial Spillover and the Threshold Effect

In this manuscript, the authors addressed an interesting issue and I found the manuscript was relatively well structured and written. I think the scientific depth and clarity of this MS can be significantly enhanced and I suggest MINOR REVISION if the authors make the following changes,
Minor Comments:
1- Please revise the title
2. The abstract doesn’t express the emphasis. It needs to be modified.
3. The authors did not mention clearly their objective in the introduction and add to the contribution of the study.
4. An overview of the other section of the article needs to be incorporated 
5- I suggest you create a section for theoretical framework in your manuscript
6- Expand the literature of introduction, literature, and discussion part more by citing relevant studies. 
6-The discussion of outcomes can further be improved by comparing the latest studies as suggested above.
7- More policy implications are needed in the conclusion section

Reviewer 4 Report

The authors are addressing a topical issue, namely the integration of digital financial inclusion and environmental regulation as factors of regional growth, using up-to-date spatial investigation tools. Overall, it is a good paper, addressing a relevant issue for regional development while preserving the environment. The discussion is compelling, and the findings are interesting. Nevertheless, I have a few concerns.

Firstly, I suggest that the authors explain their choice for an economic distance weight matrix instead of the more common spatial weights matrices, such as contiguity matrices,  geographical distance matrices, kernel matrices, etc.

Secondly, the discussion of the results can be improved. For instance, in the Lines 426-429, the statement “The direct effect of environmental regulation is significantly positive and the indirect effect is significantly negative, but the total effect is not significant. This indicates that environmental regulation has a negative spatial spillover effect. While it reduces the economic development in the region, it also reduces the economic development in surrounding areas through the radiation effect.” is disconcerting. The positive direct effect of environmental regulation implies increase (not reduction) of the economic development in the same region.You should also explain how does theradiation effect” negatively influence the neighbours. And who are the neighbours? They couldn’t be the “surrounding areas” (as you mention on line 429) since you are using an economic distance weight matrix, not a contiguity or geographical distance type spatial weights matrix.

Finally, on a minor note, the authors might consider changing the notation for the adjustment coefficient W, since it is the same notation used tor the spatial weights matrix and it is confusing. They should also mention what is the software used for estimating the spatial models.

Minor problems of grammar and style also need to be addressed. For instance, on Line 564, “This paper constructed constructs environmental regulation indicators”.

Round 2

Reviewer 2 Report

Dear Authors,

Your effort to improve the article is obvious.

Success!